# Bile Acids Transporters of Enterohepatic Circulation for Targeted Drug Delivery

**DOI:** 10.3390/molecules27092961

**Published:** 2022-05-05

**Authors:** Robin Durník, Lenka Šindlerová, Pavel Babica, Ondřej Jurček

**Affiliations:** 1Department of Biochemistry, Faculty of Science, Masaryk University, Kamenice 5, 62500 Brno, Czech Republic; rdurnik@mail.muni.cz; 2Department of Biophysics of Immune System, Institute of Biophysics, Czech Academy of Sciences, Královopolská 135, 61265 Brno, Czech Republic; sindler@ibp.cz; 3RECETOX, Faculty of Science, Masaryk University, Kotlářská 2, 61137 Brno, Czech Republic; pavel.babica@recetox.muni.cz; 4Department of Chemistry, Faculty of Science, Masaryk University, Kamenice 5, 62500 Brno, Czech Republic; 5CEITEC—Central European Institute of Technology, Masaryk University, Kamenice 5, 62500 Brno, Czech Republic; 6Department of Natural Drugs, Faculty of Pharmacy, Masaryk University, Palackého 1946/1, 61200 Brno, Czech Republic

**Keywords:** bile acid, bile salt, enterohepatic circulation, transport protein, NTCP, ASBT, IBABP, OATP, MRP, MDR, drug delivery

## Abstract

Bile acids (BAs) are important steroidal molecules with a rapidly growing span of applications across a variety of fields such as supramolecular chemistry, pharmacy, and biomedicine. This work provides a systematic review on their transport processes within the enterohepatic circulation and related processes. The focus is laid on the description of specific or less-specific BA transport proteins and their localization. Initially, the reader is provided with essential information about BAs′ properties, their systemic flow, metabolism, and functions. Later, the transport processes are described in detail and schematically illustrated, moving step by step from the liver via bile ducts to the gallbladder, small intestine, and colon; this description is accompanied by descriptions of major proteins known to be involved in BA transport. Spillage of BAs into systemic circulation and urine excretion are also discussed. Finally, the review also points out some of the less-studied areas of the enterohepatic circulation, which can be crucial for the development of BA-related drugs, prodrugs, and drug carrier systems.

## 1. Introduction to Biochemistry of Bile Acids

Bile acids (BAs) are a group of steroids formed during the enzymatic catabolism of cholesterol in the hepatocytes of the liver [1,2]. The biochemistry of BAs is a complex subject because of the great variety of chemical structures and functions of these naturally occurring steroid compounds [2,3,4]. Biliary BAs show remarkable structural diversity not just in humans, but also across various animal species—no other class of small molecules in vertebrates exhibits such a variety of chemical structures [5]. This variety can be explained by the biochemistry of cholesterol, which is a poorly water-soluble and predominantly plasma-membrane-associated steroid, and the related evolution of multiple biochemical pathways that serve for its conversion into more water-soluble, amphipathic, membranolytic molecules—BAs [2].

BAs have a rigid tetracyclic steroid skeleton, which is decorated by various numbers of reactive hydroxy groups of defined stereochemistry and by a flexible alkyl side chain bearing a carboxyl functional group (Figure 1). These unique properties predestine them for applications in biology as well as medicinal and supramolecular chemistry [6,7,8,9,10,11,12,13,14]. The BAs produced during the catabolism of cholesterol in the liver are termed primary BAs, whereas metabolites generated by bacterial enzymes in the intestine are called secondary BAs. The carboxylic acid group of BAs is conjugated with glycine and taurine in liver cells to form conjugated BAs, often referred to as bile salts [1,15].

### 1.1. Role of Bile Acids in Digestion

Bile is an aqueous solution composed of mostly bile salts, phospholipids, cholesterol, and biliary pigment proteins. Bile is concentrated and stored in the gallbladder and plays a major role in enterohepatic circulation [13]. After food intake, bile is secreted into the small intestine, where bile salts function as detergents in the digestive process by emulsifying dietary lipids and lipid soluble vitamins through the formation of mixed micelles (Figure 2). The amphiphilic character of bile salts plays a major role in the formation of micelles. BAs and bile salts are amphiphilic structures with a hydrophobic side of unsubstituted skeletal steroid structure and a hydrophilic side bearing one or more hydroxyl groups (Figure 1B), a carboxylic acid group or amide carbonyl, or ionized acidic groups of conjugated glycine or taurine [1]. Such mixed micelles are not only responsible for lipid/cholesterol solubilization in the small intestine, but also, they increase the surface area of lipids which are readily available for processing by lipases [1,13]. The conjugated BAs are largely impermeable to the cell membranes of the intestine. This allows their levels to rise in the lumen, ultimately reaching sufficient concentrations to form micelles [1,16]. Unless BAs are present in micellar form, fat-soluble vitamins (A, D, E, and K_1_) will not be absorbed, and a deficiency will occur. The mixed micelle formation also promotes cholesterol elimination. Additional function of BAs in digestive process can be in the form of their solubilization of polyvalent metals, such as iron and calcium in the duodenum, promoting their absorption [17].

After performing their role in digestion, bile salts are deconjugated by bacterial enzymes and then almost completely reabsorbed in the small intestine and colon by both active and passive mechanisms of the enterohepatic circulation [13]. It is important to note that BAs also play indirectly a role in the metabolism of carbohydrates such as glucose [18,19,20,21]. BAs have also been shown to interact and bind to dietary fiber, which influences not only their metabolism, but also the composition of gut bacteria [4,22]. As a consequence, increased fecal losses of BAs can be observed to result in an increased production of new BAs, thus regulating the level of cholesterol [23]. Bile salts were also suggested to aid protein digestion. Bile salts can adsorb to any hydrophobic domains of dietary proteins, promoting protein denaturation and making them more available to digestion by enzymes [16].

BA metabolism is closely related to the metabolism of cholesterol; altering the cholesterol level alters levels of BAs and vice versa. Cholesterol is essential for all animal life. It composes about 30% of animal cell membranes. Cholesterol is also implicated in cell signaling processes and regulation of substrate presentation and is an important chemical precursor for the synthesis of vitamin D, BAs, and all steroid hormones—i.e., adrenal gland hormones cortisol and aldosterone or sex hormones progesterone, estrogen, testosterone, and their derivatives [24].

### 1.2. Bile Acids as Signaling Molecules

For a long time, BAs were thought to function purely as compounds that would solubilize lipids and facilitate uptake of lipid-soluble vitamins through the mechanisms described above. It was not until quite recently that the scientific world started to understand that BAs are not only detergents, but also important signaling molecules that affect various receptors [25,26,27]. BAs can interact with nuclear receptors, which directly function as transcriptional regulators, as well as membrane-bound receptors, activating various signal transduction pathways. That includes also membrane-G-protein-coupled receptors, which are the largest family of cell surface proteins. Through interactions with receptors, the BAs contribute to the regulation of various metabolic and cellular processes, such as glucose homeostasis, lipid metabolism and energy expenditure, gut motility, and immune cell function. Interactions with receptors also control BA metabolism and homeostasis, which directly regulate their transport as will be described in more depth later. It is expected that the zone of recognized regulatory functions and effects of BAs will further expand in the years to come, as BA receptors have been found to be widely distributed in the human body [25,28,29,30].

Nuclear farnesoid X receptor (FXR) can serve as an example to show the complexity of the subject. FXR plays a major role in BA homeostasis, but also affects, directly or indirectly, glucose metabolism and can influence drug metabolism [19,20]. For example, in mice, activation of FXR by cholic acid was able to suppress expression of multiple genes involved in gluconeogenesis and lowered glucose levels [31]. On the other hand, ursodeoxycholic acid (UDCA) does not seem to activate this receptor [25,32]. However, short-term UDCA treatment can increase BA production in morbidly obese non-alcoholic fatty liver disease patients. This probably occurs by reducing FXR activity, which leads to hepatic cholesterol utilization, subsequently inducing cholesterol biosynthesis and neutral lipid accumulation in both the liver and visceral white adipose tissue. A decrease in FXR activation also increased hepatic triglyceride formation by increasing the amount of stearoyl-CoA desaturase—an enzyme that catalyzes the biosynthesis of monounsaturated fatty acids from saturated fatty acids [33,34].

### 1.3. Bile Acids in Humans and Mice

A large amount of experimental data is based on animal models, mainly on mice. Animal models have helped us to better understand biosynthesis, enterohepatic circulation, and metabolism of BAs. However, there are several major differences between mice and humans one must keep in mind when interpreting and extrapolating experimental data [35,36]. The first difference is in the biosynthesis of BAs. In mice, BAs are also hydroxylated at position C6, forming muricholic acids (Figure 1A), a set of BAs that are unique to mice and cannot be found in healthy humans [35,37]. Another example of interspecies differences is the origin of UDCA. UDCA is just a very small component of the BA pool in both humans and mice. In humans, UDCA is generated from chenodeoxycholic acid by the gut microbiota—hence it is a secondary BA. In mice, however, UDCA is a primary BA later transformed by cytochrome P450 to β-muricholic acid [38]. Thus, UDCA can also be detected in germ-free mice (lacking the intestinal bacteria) [36,39]. Despite the differences in BA pools, the enterohepatic circulation of BAs and the fundamentally involved transport proteins are rather similar in human and mice. Amongst the most notable differences, there is a greater reliance on organic-anion-transporting polypeptides (OATPs) for hepatic transport of conjugated BAs in mice, while other transport systems could play a more dominant role in humans. Moreover, intestinal passive absorption of BAs is higher in humans compared to mice, which is due to the larger pool of hydrophobic and glycine-conjugated BAs in humans, whereas BAs conjugate almost exclusively with taurine in mice and hence require an active transport [36,40].

There are also noteworthy differences between sexes in the biochemistry of BAs. In female mice, the BA pool is approximately 60% larger and more hydrophobic than in males. In humans, however, unlike female mice, women have a lower BA pool size compared to men [37]. As far as human studies are concerned, BA data regarding sex differences in transport proteins are very limited [41].

Apart from more “traditional” animal models, “humanized” mouse models where the liver better resembles that in humans could be used. In such cases, individual or small groups of genes in mice can be replaced by their human counterparts—however, more data are required to demonstrate the accuracy of the evaluation of outcomes [36,42]. Rodent and human ex vivo and in vitro cell cultures have also become important tools in the research of BA metabolism, transport, function, and pathophysiological effects, as well as in preclinical studies focusing on the interactions of BA transporters with drug candidates and their metabolites [43].

### 1.4. Cholestasis

Cholestasis is a term which will be used in the following chapters when describing the transport of BAs in the enterohepatic circulation. It refers to impairment of or reduction in bile flow. This pathological condition can occur when the biliary tree is blocked or when bile secretion by hepatocytes into the bile canaliculi is impaired [44,45]. If bile is accumulated and its composition is altered, hepatocytes and cholangiocytes suffer from the toxic effects caused by high concentrations of BAs. This can then progressively lead to liver fibrosis, cirrhosis, and eventually liver failure [1,36]. Altered BA metabolism caused by gene mutations, a high-fat diet, or alcohol consumption can also contribute to the development of cholestatic diseases [46].

### 1.5. Transport of Bile Acids and Enterohepatic Circulation

From a general point of view, enterohepatic circulation allows for the recycling of metabolized and non-metabolized compounds [47]. It involves the passage of substances that have been absorbed from the intestines via the portal vein to the liver and their subsequent passage via the biliary system back into the intestine [48]. The transport of BAs between organs has been mostly studied using genetically engineered mice models with targeted inactivation of genes of interest. For example, bile salt export pump (BSEP) is involved in the BA transport in the canalicular membranes of liver, and the knockout of *bsep* gene in mice resulted in intrahepatic cholestasis [18,49].

Under normal physiological conditions, BAs follow a defined route of transport within the body (Figure 3). BA flow is restricted almost exclusively to enterohepatic circulation [19]. Additionally, other endogenous and exogenous compounds undergo some degree of enterohepatic circulation, but this pathway usually represents a minor component of their tissue distribution [48]. The BA pool size is between 4–6 g for human adults and about 12–18 g of BAs is excreted in total every day by the intestine to mediate digestion. During this impressive turnover, only about 5% of BAs leave the enterohepatic circulation, passing to colon and eventually being excreted in feces [1,18]. This demonstrates the effectiveness of BA enterohepatic circulation.

The pK_a_ of the unconjugated BAs is between 5–6.5 [50] and the pH in the duodenum ranges between 3–5. Thus, if the unconjugated BAs are present in the duodenum, the carboxylic acid tends to be protonated, which makes them relatively insoluble in aqueous solution. In contrast, the pK_a_ of conjugated BAs is lower, between 1–4, thus they are present in their deprotonated form in the duodenum, which makes them significantly more water-soluble and thus able to function as emulsifiers in the digestion process [51]. However, their negative charge also prevents their passive transport through membranes and they rely mainly on membrane transporters [18,52]. A proportion of the BAs is deconjugated as they pass down the small intestine toward the ileum. Unconjugated BAs are thought to be reabsorbed passively throughout the small and large intestines, whereas the bulk of conjugated BAs are recovered mainly by ileal enterocytes by active transport [18]. Some of the most important membrane transporters and their functions will be described in the following pages.

A small number of BAs also spills into general circulation, e.g., BAs have been detected in the ovaries, heart, and brain. It is not yet fully clear what purpose they serve in these organs [18,52]. However, as mentioned above, BA-binding receptors are widely distributed across different tissues and there is increasing evidence that BAs can function as hormone-like signaling molecules in a variety of physiological processes outside of the digestive system, such as cardiac function or renal water regulation [53,54]. Renal cells are capable of using their transport proteins for the uptake of BAs from general circulation, which are then excreted into urine or, usually, are reabsorbed into enterohepatic circulation [18,52]. BAs can also be found in the pancreas—this, however, is considered to happen under pathophysiological conditions such as obstruction of the ampulla of Vater by an impacted gallstone [55].

### 1.6. Regulation of Bile Acids in Enterohepatic Circulation and Their Interactions with Nuclear Receptors

Bas, when present at high concentrations, can exert cytotoxic effects. It is possible that as a way to control and prevent the toxicity of BAs, BA levels can be regulated by nuclear receptor sensing, which is detecting and controlling their concentrations within the body. A major role is played by the nuclear receptor farnesoid X receptor (FXR), also known as bile-acid receptor or NR1H4, which is highly expressed in the liver and gut, likely because hepatocytes and bile duct cells are not only the major cells involved in the synthesis and metabolism of BAs, but are also at the highest risk of cytotoxicity [19,20]. Many of the genes encoding proteins involved in BA synthesis, metabolism, and transport are strictly controlled by BAs themselves via activation of FXR. The FXR activated by BAs heterodimerizes with RXR, functioning as a transcription factor inducing expression of the inhibitory protein SHP. SHP then suppresses the activity of the gene for cholesterol 7-α-hydroxylase (CYP7A1), the rate-controlling enzyme for cholesterol catabolism into BAs. This means that FXR affects the conversion of cholesterol to BAs in a feedback manner [1,20,56]. To keep BA levels constant in the liver, FXR can also induce synthesis of BSEP in the canalicular membrane [1]. Liver X receptor (LXR) is another nuclear receptor/transcription factor found to be involved in the autoregulation of BA levels. LXR binds to cholesterol or its metabolic products such as oxysterols and affects enzymatic conversion of cholesterol to BAs in a feedforward manner [1,56].

One receptor that could potentially protect liver from toxicity of high concentrations of BAs is pregnane X receptor (PXR). This receptor functions as a xenobiotic sensor that can induce the removal of potentially harmful chemicals from the body. PXR could induce the formation of BA conjugation enzymes and, in addition, could suppress CYP7A1 and thus decrease the BA pool. However, it is still being studied if and to what extent this receptor is activated by BAs. The current evidence suggests that PXR is only activated under extreme concentrations of lithocholic acid which even exceed levels observed in patients with severe cholestasis [25,57,58].

Vitamin D receptor (VDR) is usually activated by vitamin D. VDR is expressed mostly in osteoblasts, kidneys, and in many types of immune cells, but it can also be found in intestines and to some degree in the liver. VDR plays a central role in the biological function of vitamin D—regulation of calcium homeostasis, cellular proliferation, and immunity. Activation of VDR by secondary BAs was found to induce BA-metabolizing cytochrome P450 (CYP3A4) that detoxifies BAs in the intestine. Similarly to PXR, lithocholic acid showed the ability to activate VDR and thus influence the regulation of genes involved in BA metabolism and transport [25,59,60].

## 2. Bile Acid Transport Proteins within Enterohepatic Circulation

### 2.1. Hepatocellular Transport

The enterohepatic circulation of BAs involves their transport in the liver from portal blood (liver sinusoids) into bile canaliculi, which occurs across the hepatocytes and against a concentration gradient [50]. Moreover, new BAs are also synthesized in hepatocytes. Thus, hepatocellular secretion of BAs into bile canaliculi is a major mechanism for both hepatic bile formation and ongoing enterohepatic BA circulation [1,61]. This transport process is governed by the polarized expression of distinct active transport systems localized at the basolateral (sinusoidal) and apical (canalicular) surface domains of hepatocytes (Figure 4) [52,61]. The liver is an inhomogeneous system of hepatocytes where proteins vary across its entire volume [62].

An overview of hepatocellular transport can be seen in Figure 4. In plasma, BAs circulate mainly bound to albumin, but they are also known to bind to high-density lipoproteins. Not only protein-bound BAs, also referred to as protein-conjugated BAs, but also protein-unconjugated BAs returning in portal blood from the intestinal lumen are taken up by the hepatocytes with high efficiency via Na^+^/taurocholate-co-transporting polypeptide (NTCP) and organic-anion-transporting polypeptides (OATPs) [1,18,52]. Simple dissociation of BAs from serum albumin is slow. The dissociation thus very likely proceeds due to conformational changes in the structure of the complex upon the contact of albumin with the basolateral membrane of the hepatocytes. This hypothesis has been confirmed by studies performed on rats [63], but experimental proof from human studies is still lacking [1,64].

Efflux of BAs from the basolateral membrane is negligible under normal physiological conditions but can occur in cholestasis. BAs would then be returned to blood by the multi-drug-resistance-associated protein 3 or 4 (MRP3 or MRP4) [1,52]. Unconjugated BAs are weak acids that are uncharged at physiological pH in plasma. This means they can also travel through membranes by passive diffusion—this, however, is not very efficient and specific transport proteins are used instead. Entry of unconjugated BAs into cells by simple diffusion can, however, occur if there is a steep concentration gradient [18,64].

In general, the intracellular transport of BAs across the hepatocyte is not well understood. It is possible that it is facilitated by either intracellular trafficking or by vesicle-mediated transport. Under normal physiological conditions, a large portion of BAs is bound to intracellular binding proteins and diffuses into the canalicular membrane in this form. When presented with higher BA loads, an increasing amount is transported by organelles such as the endoplasmic reticulum or Golgi apparatus [52,64].

The next step, BA canalicular secretion, determines at what rate the bile is formed. After crossing the hepatocyte, BAs are exported into the canaliculus by BSEP. Phosphatidylcholine (PC) is also transported from the inner part of the apical membrane by multi-drug-resistance protein 3, MDR3 and then is mixed with BAs secreted by BSEP. MRP2 is capable of transporting BAs into the bile canaliculus, but it cannot transport non-sulfated monovalent BAs. Aquaporins (AQP, water channels) secrete water into bile [1,52,64].

BAs that escape clearance by hepatocytes are filtered at the glomerulus from plasma into urine. Under normal conditions, the majority of BAs are reabsorbed, and urinary BA levels are therefore minimal. Under cholestasis, renal excretion of BAs can become an important alternative pathway which helps to maintain homeostasis [64]. For example, in a study performed on six women with primary biliary cirrhosis and varying degrees of cholestasis, urinary BA levels markedly increased in proportion to BA serum levels [65].

#### 2.1.1. Basolateral Hepatocellular Transport

##### Na^+^/Taurocholate-Co-Transporting Protein (NTCP)

In general, BAs are transported into hepatocytes by both Na^+^-dependent and Na^+^-independent mechanisms. NTCP is a member of the solute carrier (SLC) group of membrane transport proteins, which is comprised of more than 450 members categorized into 65 superfamilies [66]. Specifically, NTCP belongs to SLC family 10, and it is encoded by gene *slc10a1* [67]. NTCP, also known as liver bile acid transporter (LBAT), mediates Na^+^-dependent uptake of BAs. The Na^+^-dependent method of transport is considered to be responsible for more than 80% of conjugated taurocholate uptake but also less than 50% of unconjugated cholate uptake [62,64,68]. NTCP is not only responsible for the transport of BAs but also of other organic anions and compounds—for example, estrogen conjugates, thyroid hormones, or drugs that are covalently bound with taurocholate [62,64].

The force which allows for the transport is provided by the inwardly directed Na^+^ gradient. This gradient is maintained by basolateral Na^+^/K^+^ ATPase, the outward diffusions of K^+^, and the cell’s natural intracellular potential. The electronic transport of Na+/taurocholate seems to happen in stoichiometry of at least 2:1 [64,69]. The structure of NTCP shows 77% amino acid homology between rats and humans [62].

It was reported that NTCP is localized only at the basolateral membrane of hepatocytes [64]. However, at least in rats, NTCP is also expressed at the apical membrane of pancreatic acinar cells [69,70].

NTCP deficiency in some patients indicates that there are different ways and mechanisms through which BAs can enter the hepatocytes, as some patients studied with NTCP deficiency had no clinical signs of liver disfunction [71,72]. However, the transport capacity of these alternative mechanisms, which will be described below, seems to be insufficient to fully compensate for the NTCP deficiency [71]. One could, therefore, still expect a discovery of new transport proteins in the future.

##### Organic-Anion-Transporting Polypeptide (OATP)

Na^+^-independent BA transport is mediated by OATPs. This method of transport is currently considered quantitatively less significant than the Na^+^-dependent method manifested by NTCP [62,64]. Na+-independent processes can be found in lower vertebrates which lack the Na+-dependent mechanisms—this suggests that Na^+^-dependent transporters evolved more recently [1,73].

OATP belongs to SLC family 21, encoded by *slco* genes. Eleven human OATPs have been identified and classified into six subfamilies (1–6). Due to gene duplication and divergence, different orthologs can be found in humans (OATP) and rodents (Oatp): OATP1A2 has five rodent orthologues: Oatp1a1, Oatp1a3 (in rats only), Oatp1a4, Oatp1a5, and Oatp1a6. OATP1B1 and OATP1B3 have a single rodent ortholog, Oatp1b2. The other OATPs and their rodent orthologs are OATP1C1 (Oatp1c1), OATP2A1 (Oatp2a1), OATP2B1 (Oatp2b1), OATP3A1 (Oatp3a1), OATP4A1 (Oatp4a1), OATP4C1 (Oatp4c1), OATP5A1, and OATP6A1 (Oatp6b1, Oatp6c1 and Oatp6d1) [74].

Members of the OATP family are multispecific transporters responsible primarily for cellular uptake of a broad range of large, fairly hydrophobic, and mostly anionic substrates, including bile acids, steroid conjugates, and numerous xenobiotics. Apart from both conjugated and unconjugated BAs, OATPs have wide substrate specificity which, for example, includes bilirubin and many drugs used clinically such as statins, macrolide antibiotics, and antihistamines [62,64,74]. OATPs are expressed in a tissue- and cell-type-specific manner in numerous epithelia and organs across the body, including the brain, lungs, heart, liver, intestine, kidney, testes, and others [1,73,75]. OATP1B1, OATP1B3, and OATP2B1 are expressed in the basolateral membranes of hepatocytes, with OATP1B1 and OATP1B3 found to contribute to hepatocellular uptake of various BAs [61,76].

Na^+^-independent BA transport is thought to be responsible for 20–25% of total uptake from portal blood, as shown on rat hepatocytes. This transport process is higher for unconjugated Bas, where it seems to be responsible for most of the uptake. Transport of unconjugated BAs can, however, occur by passive diffusion as well [52,62]. The structure of a BA has an overall impact on whether it will be transported by Na^+^-dependent or independent mechanism, e.g., BAs with a hydroxy group at carbon position C6 prefer Na^+^-dependent uptake [62,77].

The transport mechanism appears to be largely mediated by exchange with intracellular anions such as HCO_3_^−^ or deprotonated glutathione [62,64].

##### Multi-Drug-Resistance-Associated Protein 1, 3, and 4 (MRP1, MRP3, and MRP4)

In the basolateral membrane of hepatocytes, BA uptake is the predominant path of transport under normal physiological conditions. Increased efflux of BAs can, however, occur under cholestatic conditions. Efflux transport systems are manifested by the MRP subfamily, which belongs to the class of ATP-binding cassette (ABC) transport proteins [52,64,78]. The MRP subfamily is thought to be comprised of six members (MRP 1–6), some of which are expressed in the liver to a larger or smaller extent. In general, MRPs are present mostly in other tissues with a barrier function—lungs, intestines, or the blood-brain barrier [64,78]. MRP3 (encoded by *abcc*3 gene) and MRP2 (encoded by *abcc*2 gene) are significantly present in the liver, where MRP3 is located at the basolateral surface while MRP2 is located at the canalicular surface (more on this protein in the next section). Function of MRP3 and MRP2 seems to be bound together—levels of MRP3 increase whereas MRP2 decreases under cholestasis conditions. This seems to protect the liver from toxicity of high concentrations of BAs [78,79]. BAs were also reported to be the substrates for MRP4 (encoded by *abcc*4 gene), which seems to contribute to the elimination of BAs from the hepatocyte to sinusoidal blood. MRP3 is also expressed the most during cholestasis [78,80]. Another transporter is MRP1, which is barely detectable in the human liver, but can be expressed more under specific conditions, e.g., in proliferating hepatocytes and liver cancer cell lines [79].

During cholestasis, there is a significantly increased expression of MRP1 and MRP3, which are responsible for the transport and elimination of sulphated and glucuronidated BAs as elimination products excreted in urine [64].

#### 2.1.2. Apical Hepatocellular Transport

##### Bile Salt Export Protein (BSEP)

Canalicular secretion at the apical side of hepatocytes is often regarded as the rate controlling step of BA transport through the hepatocytes. BSEP is thought to be the most dominant transporter of the canalicular membrane responsible for the excretion of BAs into bile, which runs against a steep concentration gradient. BSEP, also known as a “sister of P-glycoprotein”, is encoded by the *abcb*11 gene, which belongs to the ABC superfamily of transporters [49,52,78,81].

Thus far, it is understood that there is no backup transporter for the canalicular export of BAs in humans, as inactivation of BSEP leads to liver damage. Progressive familial intrahepatic cholestasis type 2 could serve as an example—here, the deficiency of BSEP is inherited and leads to lethal progressive familial intrahepatic cholestasis [78,82]. On the other hand, in mice studies, BSEP knockout showed only mild and non-progressive cholestasis, which suggests the existence of another canalicular BA transport system in mice [49].

BSEP mainly transports primary and secondary BA salts as well as UDCA and its conjugates. As an example, chenodeoxycholic acid seems to be eliminated by BSEP at a faster rate compared to cholic acid. This might be because of the fact that chenodeoxycholic acid and its conjugates are potentially more toxic for hepatocytes even at lower concentrations, hence faster elimination is desirable. UDCA showed the lowest toxicity in this study done on mice [81,83].

Although BSEP does not seem to transport drugs or drug metabolites, some compounds may act as competitive BSEP inhibitors. This can lead to cholestasis and potential liver injury. Thus, many pharmaceutical companies incorporated BSEP inhibition into their in vitro screening panels in preclinical studies [84]. However, such in vitro screening of the parental drug can fail to reveal hepatoxicity due to the BSEP-inhibiting effects of intrahepatically produced drug metabolites [78,85,86]. BSEP inhibition can also occur during pregnancy and contribute to disorders such as intrahepatic cholestasis of pregnancy [87,88].

##### Multi-Drug-Resistant Protein 3 (MDR3)

MDR proteins, or P-glycoproteins, also belong to the superfamily of ABC transporters, which represents ATP-dependent efflux pumps with broad substrate specificity. MDR1, or P-glycoprotein, (encoded by the *abcb*1 gene) is known to affect the distribution of various drugs in the body. However, it is expressed only at relatively low levels in normal healthy livers. The canalicular membrane of hepatocytes contains MDR3 (encoded by the *abcb*4 gene), which is structurally similar to MDR1, as it shares 77% of its amino acid composition, but it is functionally a different transporter. MDR3 is a PC transporter which is capable of translocating phospholipids to the outer side of the membrane lipid bilayer. In contrast to MDR1, MDR3 does not seem to be able to transport drugs directly [78,89]. Nevertheless, MDR3 can be inhibited by drugs’ activity, e.g., itraconazole, posaconazole, or ketoconazole. Their effect could lead to cholestatic liver injury because the phospholipid excretion into bile is slowed down; thus, the bile becomes more toxic since mixed micelle formation is prevented. Azoles can also inhibit BSEP activity, which can further contribute to cholestatic injury [78,90].

##### Multi-Drug-Resistance-Associated Protein 2 (MRP2)

In contrast to MRP3 and MRP4, MRP2 (encoded by *abcc*2) is directed towards the canalicular system. It is responsible for the transport of a wide range of substrates—including bilirubin conjugates, drug metabolites, unconjugated drugs and glucuronide, and sulfate or glutathione conjugates—including conjugates of BAs or drugs [1,52,64,78].

##### Aquaporins (AQP)

Bile consists of about 95% of water. The AQP family of membrane proteins functioning as water channels is expressed both at the apical and basolateral plasma membrane and is involved in water transport. Specifically, AQP0, AQP8, and AQP9 are expressed in hepatocytes [1,91].

### 2.2. Cholangiocellular Transport, Transport to Small Intestine

For a long time, it was assumed that biliary components such as BAs do not interact with bile ducts after canalicular secretion. Later, it was discovered that cholangiocytes, a variable group of epithelial cells which form the biliary epithelium lining the system of bile ducts (Figure 5B), can further modify the composition and volume of canalicular bile secreted by hepatocytes, thus forming so-called ductal bile. Formation of ductal bile is the result of transport-protein-generated movement of solutes and osmotically driven water molecules [1,91,92,93,94]. Modification of the bile involves secretion of water, Cl^−^, and HCO_3_^−^ into the duct and removal of glucose and amino acids. The return of BAs back to the hepatocytes via the cholehepatic shunt pathway could also be, among other things, considered a modification mechanism of bile composition [93]. Bile acid flow is also regulated by a wide range of factors including the hormone secretin and the peptide bombesin. Bile duct secretion and transport are hence much less constant in contrast to hepatocytes [1,95]. Bile-duct-, peribiliary-gland-, and gallbladder-secreted mucin mainly provides cytoprotection to the epithelium, whereas transepithelial fluid prevents the stagnation of BAs along the biliary tract in its entirety [94].

The biliary tract can be divided into the intrahepatic and extrahepatic sections (Figure 5). The intrahepatic section’s structure resembles the structure of a tree—it is a network of bile ductules and ducts that extends from the canals of Hering, which represent the link between hepatocyte canaliculi and the intrahepatic biliary tree all the way to the hepatic ducts. Hepatic ducts then belong to the extrahepatic section, which is comprised of the cystic duct, the gallbladder, and the common bile duct. After being modified (or stored during the inter-digestive phase in the gallbladder), bile eventually reaches the duodenum through this complex system [1,95,96].

Regarding BA transport, the peribiliary plexus (Figure 5A) is an important component of the biliary tract. Through the plexus, BAs can be cycled back to the hepatocytes after being absorbed by cholangiocytes. After being cycled to the hepatocytes, BAs are then once again excreted across the canalicular membrane. This process is called the cholehepatic shunt pathway and it allows for an increased BA biliary transit time, increased bile flow for each BA molecule secreted, and increased biliary lipid secretion [97]. For example, in the case of cholestasis, the cholehepatic shunt pathway appears to be an important pathophysiological response which creates an alternative method for the absorption and recycling of BAs, which then in turn compensate for the ileal absorption [52,92]. As far as physiological relevance is concerned, cholangiocyte BA uptake might be more important for signaling—triggering adaptive intracellular responses, e.g., regulation of cholangiocyte secretory activity. This was suggested by the fact that the cholangiocellular apical sodium-dependent bile acid transporter (ASBT) does not appear to be a high-throughput system for physiological BAs [52,98]. An overview of specific proteins responsible for this transport can be seen in Figure 5.

Apical transport from the system of bile ducts into the cholangiocytes is mediated by a similar set of proteins that can be found in the terminal ileum, which will be discussed in a later section. BA uptake is managed by ASBT. AQPs are present at both membranes. Bicarbonate anions are secreted by the chloride/bicarbonate ion exchanger (CBIE)—this helps to generate bile flow as well as makes cholangiocytes less likely to undergo apoptosis. Absorbed BAs are then moved across the basolateral membrane by MRP3, organic solute transporter α/β (OST), or by the truncated form of ASBT (tASBT) [1,52,93,99]. Another protein possibly involved in BA transport is OATP1A2, which is expressed throughout different tissues, including both apical and basolateral membranes of cholangiocytes [74].

#### 2.2.1. Apical Cholangiocellular Transport

##### Apical Sodium-Dependent Bile Acid Transporter (ASBT)

ASBT is a protein that was first identified in the renal tubule and is mainly present in the small intestine, thus it is also known as ileal bile acid transporter (IBAT). ASBT is encoded by the *slc10a2* gene, which represents another member of SLC family 10, similarly to NTCP. The role of ASBT in bile ducts might not be as obvious as in the small intestine or kidney. Uptake of BAs is sodium-dependent and activates cholehepatic shunting as described previously [1,92,97]. ASBT has a substrate specificity for unconjugated and conjugated BAs as concluded from studies performed on human ileal and renal transporters [92,100].

It was proposed that the interaction of ASBT with BAs could also lead to a certain protection mechanism, such as stabilization of the HCO_3_^−^ apical barrier, which might keep certain BA conjugates in a deprotonated and membrane impermeable form, thus protecting the cholangiocytes against potential BA toxicity at high concentrations [101].

##### Aquaporin (AQP)

Cholangiocytes were found to express AQP1 at both the apical and basolateral membranes (Figure 5C). Secretin could lead to promotion of insertion of this protein into the apical membrane but had no effect on the basolateral membrane [1]. Expression of AQPs, such as AQP1, has been well documented in rat cholangiocytes, but remains rather unclear in human models. One major difference between rats and humans/mice is that rats do not have gallbladders, suggesting significant differences in the function of cholangiocytes [91,102]. Other AQPs, such as AQP3 and AQP8, were also identified in cholangiocytes of mice [102].

##### Chloride/Bicarbonate Ion Exchanger (CBIE)

CBIE, or anion exchange protein 2, is a Na^+^-independent Cl^−^/HCO_3_^−^ exchanger, which is encoded by *slc4a2* (*ae2*) gene of SLC family 4 [103]. CBIE in cholangiocytes is considered to be a major protein responsible for HCO_3_^−^ secretion into bile [1,95,103]. It can be activated by secretin through a secretin receptor that is also expressed in cholangiocytes [93]. Penetration of BA conjugates into cholangiocytes is pH-dependent, therefore their toxicity is also determined by extracellular pH. Alkalization of bile by HCO_3_^−^, also known as the HCO_3_^−^ umbrella, might be a key protective mechanism [104,105]. Certain genetic variants of this protein have also been associated with improved prognosis of biliary cirrhosis [104,105].

#### 2.2.2. Basolateral Cholangiocellular Transport

##### Multi-Drug-Resistance-Associated Protein 3 (MRP3)

As can be seen from the previous sections, MRP3 can be found in various parts of enterohepatic circulation. It can also transport BAs across the basolateral membrane of cholangiocytes into the peribiliary plexus. Levels of MRP3 can be increased with various forms of cholestasis [1,64,106].

##### Aquaporins (AQPs)

In addition to AQP1, which is found also in the apical membrane, transport of water across basolateral membrane is also facilitated by AQP4. This allows maintenance of the isosmolar status of the cells during bile formation [91].

##### Organic Solute Transporter α/β (OST)

OST is a transporter localized in the basolateral membrane of different epithelial cells involved in the enterohepatic circulation. It advocates for Na+-independent efflux of BAs across the basolateral membrane. OST is a heterodimer and requires coexpression of both OSTα and OSTβ proteins, which are encoded by genes from the SLC family 51, *slc51a* and *slc51b* [1,92,103,107]. Even outside the cholangiocytes, these two subunits of the transporter are coexpressed in tissues that express ASBT. ASBT and OST have overlapping substrate specificities over a large variety of sterols [108]. OST transport was found to be independent of Na^+^, K^+^, H^+^, and Cl^−^ gradients and ATP levels; hence, it seems to transport substrates by simply allowing them to passively move according to their concentration gradient [108,109].

##### Truncated ASBT (tASBT)

The spliced form of ASBT or truncated ASBT which, in contrast to ASBT, can function as a basolateral BA transporter. Its specific contribution to the efflux of BAs, however, seems to be minor or uncertain [52,110].

### 2.3. Storage in Gallbladder

Gallbladder epithelium has also an ability to modify the composition of bile. Gallbladder epithelial cells and cholangiocytes share many functions and are similar in the composition of transport proteins, yet their physiological roles are quite different [1,64,92]. The gallbladder serves as a storage for bile during interdigestive phases. Increased concentration of bile is achieved by the transport of Na^+^, Cl^−^, and HCO_3_^−^, leading to passive transport of water—this involves AQPs at both the apical and basolateral membranes [1]. BAs stimulate the secretion of mucin and Cl^−^ mostly in a Na^+^-dependent manner achieved by interaction with ASBT at the apical membrane. Na^+^-independent secretion could be related to the interaction of BAs with OATPs, such as OATP1A2, which are, however, expressed only in low levels and may probably occur across both the apical and basolateral membranes. Mucin secretion serves as a defense mechanism that protects the epithelium surface. Cl^−^ secretion in turn promotes secretion of HCO_3_^−^ and water—this is most likely the mechanism that aids the gallbladder in emptying [1,94]. MRP2 was also identified in the apical membrane, similarly to MRP3, which was located in the basolateral membrane—there is evidence for the capacity of the gallbladder to secrete xenobiotics and endogenous anionic conjugates via these proteins [64,111].

### 2.4. Transport in the Small Intestine

After meal ingestion, but also following olfactory or visual sensing of food, BAs are excreted from the gallbladder into the duodenum, where they participate in the absorption of lipids and lipid-soluble vitamins by acting as biosurfactants [1,112,113]. Specifically, they mainly interact with the predominant dietary lipids—triacylglycerols, phospholipids, and cholesterol esters—breaking down large lipid droplets into smaller ones, which significantly increases their surface area, making them more accessible to lipases [112,114]. This process can only occur if BAs are above a critical micelle concentration (CMC). Conjugated BAs are strong acids with pK_a_ value of 2 for taurine and 4 for glycine conjugates, while the intestinal pH normally exceeds 5, making the BA salts fully ionized. This prevents them from penetrating the cell membranes of enterocytes and thus enables reaching the CMC. Moreover, molecules of BA salts are too large to pass across paracellular junctions [1,115].

BAs are not actually essential to the digestion of dietary fats but are necessary for their effective or complete absorption [1,116,117]. Detailed steps leading to intestinal fat absorption are not yet completely understood, but we know that different pathways are taken by different types of lipids [114,118]. Lipases hydrolyze the lipids into fatty acids and respective monoacylglycerols, lysophospholipids, and free cholesterol [112,117]. Hydrolyzed products are taken up by enterocytes, which then resynthesize lipids and package them into chylomicrons for secretion. Subsequent hydrolyzation of these lipids can occur during blood circulation, which provides energy in the form of free fatty acids. Remaining lipidic particles are removed from circulation by the liver [114]. Once the BAs perform their function in this process, approximately 95% of them are reabsorbed and shuttled across the enterocytes to the basolateral membrane, effluxed into the portal vein, and continue towards liver within the enterohepatic circulation [1,112,119]. This happens by a combination of absorption in the duodenum and jejunum, active transport in the distal ileum, and passive absorption in the colon [119]. Although unconjugated BAs with higher pK_a_ values can be freely transported by passive diffusion, this only accounts for a small fraction of their transport [112]. The terminal ileum (Figure 6) is the primary site of BA reabsorption in humans [119,120]. In the proximal small intestine, both passive and active transport were suggested to take place [119]. The contribution of jejunal uptake is still debated, but it seems to be possible based on the evidence coming mostly from in vivo rat studies [119,121,122]. Similar to the jejunum, the duodenum is also hypothesized to be capable of BA transport, but there is a lack of studies on humans [119,123].

Due to the lack of data on the other intestinal pathways, only the ileal transport will be described in more depth. Its overview can be seen in Figure 6C. Although unconjugated BAs are not ionized at the pH of the lumen, which in turn allows them to be passively absorbed, the majority of BAs are effectively transported across the apical brush border membrane by active ASBT transporters [1,119,124]. Intracellular BA transport is mediated by cytosolic intestinal-BA-binding protein (IBABP) [50]. Regardless of their uptake by passive or active transport, BAs ionize at the neutral pH of the cytosolic compartment and thus require an efflux carrier for further basolateral movement [119]. This can occur through α/β (OST), MRP3, and/or tASBT transporters [1].

#### 2.4.1. Apical Transport in Ileal Enterocyte

##### Apical Sodium-Dependent Bile Acid Transporter (ASBT)

ASBT is a protein which was already introduced in connection to cholangiocytes. Nevertheless, regarding the small intestine and enterocytes, there is more data available. This is partly because the studies in this area are motivated by ASBT’s important role in drug or drug-carrier delivery. Secondly, ASBT is expressed much more in the terminal ileum than in the biliary epithelium [1,92,125,126]. Its pharmaceutical significance will also be discussed in this section.

The ASBT is responsible for most BA absorption from the gut lumen. The transport proceeds in an electrogenic and Na+-dependent manner with a stoichiometry of sodium to BA of 2:1. In other words, BA uptake is driven by the negative intracellular potential and Na^+^/K^+^-ATPase, which maintains the Na^+^-concentration gradient. The binding of the substrates in the extracellular pocket along with Na^+^ binding drives the conformational change between the outward- and inward-facing states of ASBT, opening the cytoplasmic channel to traverse BAs. Thus far, the crystal structure of mammalian ASBT has not been determined, making the understanding of the exact mechanisms of substrate binding and transport at the molecular level limited [125,126,127]. However, crystal structures of bacterial ASBT homologs from *Neisseria meningitidis* and *Yersinia frederiksenii* have been reported [128].

Concerning substrate specificity, ASBT transports hydrophilic conjugated BAs more efficiently than hydrophobic unconjugated BAs. However, dihydroxy BAs seem to have a higher affinity for ASBT-mediated transport than trihydroxy BA [126]. Although the usual substrate for membrane transporters is usually confined to molecular sizes below 1000 Da, ASBT was shown to take up macromolecular substrates conjugated with Bas, as was demonstrated on deoxycholic-acid-based oligomeric substrates and low-molecular-weight heparin–tetra deoxycholic acid conjugates [126,127].

Oral delivery of drugs is a challenging field. The digestive system is naturally designed to degrade proteins and peptides chemically and enzymatically into shorter peptides and amino acids and lipids (triglycerides) into fatty acids, monoglycerides, and free glycerol prior to absorption. For systemic drug absorption, the drug molecule must pass through the gastrointestinal tract intact [129]. Therefore, ASBT is an interesting target for the treatment of BA-related diseases and for efficient prodrug delivery into the organs of enterohepatic circulation. In general, the drug can be attached to Bas, which then act as transporters or vector molecules [13]. The BA–drug conjugate is then recognized by ASBT as a substrate and is absorbed into the enterocyte. Since low oral bioavailability often limits clinical application of drug candidates, attachment to a transporter which can enhance intestinal absorption is a promising strategy. A different approach to increasing drug absorption involves so-called “substrate mimicry”, in which the three-dimensional drug structure resembles that of natural substrates [125,126]. Despite efforts in both academic and pharmaceutical laboratories in the area of ASBT delivery, there are no prodrugs which have progressed to the market yet. However, drug discovery studies in this direction are still ongoing [126,130,131]. Because of ASBT and NTCP substrate overlap, ASBT-targeted prodrugs could also be viewed as potential substrates for NTCP [126].

Unsurprisingly, defects in ASBT transport activity or regulation may result in a variety of gastrointestinal disorders, such as Crohn’s disease. Mutation in ASBT gene *slc10a2* can cause BA malabsorption, chronic diarrhea, fat-soluble vitamin malabsorption, and reduced plasma cholesterol levels [126]. On the other hand, controlled drug-conducted inhibition of ASBT can also be desirable. It was shown in animals, but also in human models, that plasma cholesterol levels are significantly lowered by specific inhibitors of ASBT [126,128,132]. ASBT inhibition could also be used in the treatment of certain pathological conditions [126,132,133], to mediate removal of neurotoxic BAs and ammonia from blood, or to prevent liver failure in liver cirrhotic patients [126,134]. On the other hand, the inhibition of ASBT results in greater passage of BAs into the colon, where physiological concentrations are generally low. This may cause diarrhea and abdominal pain. Therefore, careful and proper dosage of such drugs would be crucial [126].

#### 2.4.2. Intracellular Transport in Ileal Enterocyte

##### Intestinal-Bile-Acid-Binding Protein (IBABP)

IBABP, also known as fatty acid binding protein 6, is encoded by the *fabp6* gene. IBABP is primarily expressed by enterocytes, where it is responsible for the shuttling of BAs across the cytosol to the basolateral membrane. Much lower levels of expression of this protein are also shown in cholangiocytes [1,125]. One molecule of IBABP is capable of binding two BA molecules. This protein functions as an additional mechanism through which BA levels are regulated within the enterocyte—its expression is upregulated through FXR activation, while ASBT expression is downregulated. IBABP belongs to the family of intracellular lipid-binding proteins that is also capable of binding fatty acids, retinoids, and cholesterol [1,135]. Currently, the exact binding features of IBABP are not well understood [126].

Proposed transport mechanisms through ileal enterocytes are illustrated in Figure 7. Membrane-based intestinal transporters are confined to the transport of only small molecular substrates such as BAs or small BA–drug conjugates by the means of non-vesicular transport (Figure 7A) [126,127]. In the case of BA-containing macromolecular substrates or nanoparticles, different strategies of absorption and transport across the enterocytes were proposed. Low-molecular-weight heparin–tetra deoxycholic acid conjugate formed a complex with ASBT, which was rapidly taken up into cells in vesicles. The substrate could then dissociate from the ASBT (the ASBT was recycled back into the membrane), leaving the vesicular transport while binding to cytoplasmic IBABP, leading to its exocytosis at the basolateral side and preventing its entry into lysosomes (Figure 7B) [126,127]. Another mechanism can be observed for BA-containing nanocomplexes. The nanoparticle composed of orally delivered forms of heparin, protamine, and deoxycholic acid is too large for ASBT [136]. Hence, it could not be absorbed through its internal pore, but instead it resulted in an uptake of the whole nanoparticle–multiple ASBT complex, similar to receptor-mediated endocytosis. Here, however, no evidence of exocytosis was observed. The absorbed nanocomplex remained enclosed in the epithelial cell until cell elimination through natural turnover (Figure 7C) [126,136].

#### 2.4.3. Basolateral Transport in Ileal Enterocyte

##### Organic Solute Transporter α/β (OST)

This protein was already introduced in Section 2.2 on basolateral cholangiocellular transport. In the small intestine, this protein is mostly expressed in the terminal ileum, where it plays a major role in mediating the efflux across the basolateral membrane into portal blood which allows BAs to return to the liver [119,137,138,139,140].

##### Multi-Drug-Resistance-Associated Protein 3 (MRP3)

This protein was previously described regarding its function on both hepatocytes and cholangiocytes (2.1 and 2.2). In the intestine, it was assumed that MRP3 plays a significant role in the enterohepatic circulation of BAs. However, it was later discovered that its contribution to intestinal absorption is only minor, as was shown in animal studies [141,142,143]. Nevertheless, its contribution might increase during the loss of transporters such as OST [137,144].

##### Truncated ASBT (tASBT)

Lastly, tASBT is a special form of ASBT with different transporting mechanisms. Its expression was previously mentioned regarding cholangiocytes. It functions as BA efflux protein in the ASBT-expressing epithelium [110,126].

### 2.5. Transport in Colon, Fecal Excretion

Most BAs encounter low populations of bacteria in the small intestine and return to the liver relatively unchanged. However, the remaining 5% of BAs that avoid active and passive transport in the small intestine enter the colon, which is home to an ecosystem of complex microbial flora. Here, the deconjugation of conjugated BAs via the action of bacterial enzymes occurs [1,145,146]. Fecal excretion or formation of secondary BAs by bacterial enzymes follows—the conversion of primary BAs into secondary BAs is achieved by the removal of the hydroxyl group from the carbon C-7 by 7α-hydroxylase. The secondary BAs can be then reabsorbed passively across the wall of the colon into the blood supply [1,119,145]. The deconjugation also raises the pK_a_ of BAs, making passive absorption more likely. Although this is not a major site of transport, colonic absorption may play an important role in cases of BA malabsorption in other parts of enterohepatic circulation [1,145,147].

## 3. Conclusions

Bile acids (BAs) are no longer believed to be just simple molecules involved in digestion. The field of chemistry and biochemistry of BAs has advanced greatly during recent decades, possibly due to their growing role in pharmacy and biomedicine. Quite recently, new discoveries were made about their role as hormone-like signaling molecules. In this review, we summarize up-to-date knowledge obtained about BA transport proteins, their presence, and processes involved in the enterohepatic circulation, which are complemented by graphical illustrations. Moreover, BA transport through the liver, biliary ducts, gallbladder, and intestines are described as well as BA urinary output and spillage into systemic circulation. Major transport proteins such as NTCP, OATP, ASBT, and IBABP involved in BA circulation are described in depth. This overview sets to provide an elemental understanding of the complexity of BA biochemistry, which is crucial in the development of BA-derived drugs, prodrugs, or drug carrier systems.

Still, much remains unknown in this field. The crystal structure of mammalian ASBT has not been solved to date, which also represents one of the limiting factors in fully understanding BA transport in the small intestine. It has also been proposed that there is still an undiscovered BA transport mechanism taking place in hepatocytes which could additionally contribute to the understanding of processes involved and ultimately lead to novel approaches in drug design. The transport of BAs in cholangiocytes is not nearly as well studied as transport in enterocytes and hepatocytes. Possible mechanisms of transport of the nanoparticle form of prodrugs have been proposed for enterocytes; however, such studies are missing for hepatocytes. Additionally, the understanding of the transport of nanoparticles from the small intestine into the liver is low.

Finally, we anticipate further development and exciting discoveries in the chemistry and biochemistry of BAs, their pathways, and of their role in the enterohepatic circulation and beyond in the years to come, to which we also wish to contribute.

## 4. Methodology

This paper represents a narrative review which consolidates previous work in a descriptive and nonquantitative manner. The literature search was conducted using scientific literature databases Web of Science^TM^ (Clarivate, Philadelphia, PA, USA), Scopus^®^ (Elsevier, Amsterdam, The Netherlands), and PubMed^®^ (U.S. National Library of Medicine, Bethesda, MD, USA). Major search terms (title, abstract, keywords) included “Bile acid”, “Bile Salt”, “Enterohepatic circulation”, “NTCP”, “ASBT”, “IBABP”, “OATP”, “MRP”, “MDR” (abbreviated or full names), “Drug delivery”, “Drug carrier”, and their combinations. The search results were manually curated and organized using Zotero (Corporation for Digital Scholarship, George Mason University, VA). Titles and abstracts of the studies were screened and selected according to their relevance. Full texts were then reviewed, and highly relevant papers (e.g., review articles) indicated further sources (e.g., textbooks, monographs) not found by searching the databases. Relevant information was compiled into a comprehensive narrative synthesis of previously published work (1) to address the key components and processes involved in enterohepatic circulation of BAs, (2) to provide new insights emphasizing previously nonobvious connections between enterohepatic circulation and BA-related drugs, prodrugs, and drug carrier systems.

## Figures and Tables

**Figure 1 molecules-27-02961-f001:**
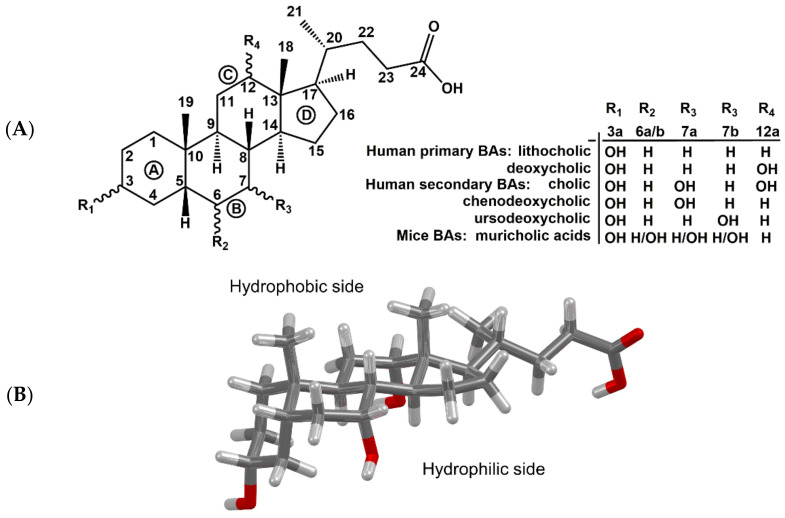
(**A**) Atom numbering and alphabetical ring labeling of steroid skeleton of selected bile acids. (**B**) Spatial arrangement of cholic acid molecule (hydroxyls in position 3α, 7α, and 12α).

**Figure 2 molecules-27-02961-f002:**
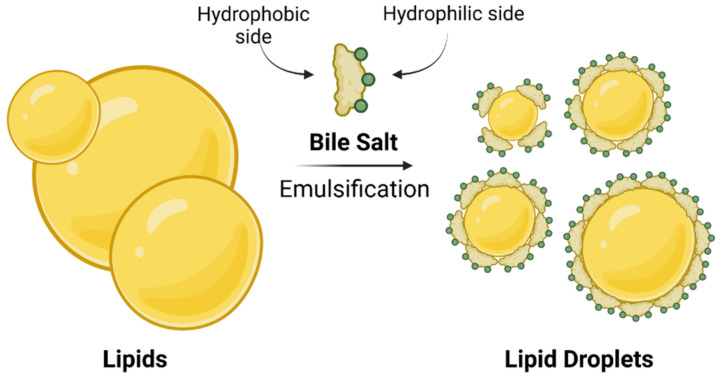
Simplified model of lipid solubilization in aqueous environment. Created with BioRender.com.

**Figure 3 molecules-27-02961-f003:**
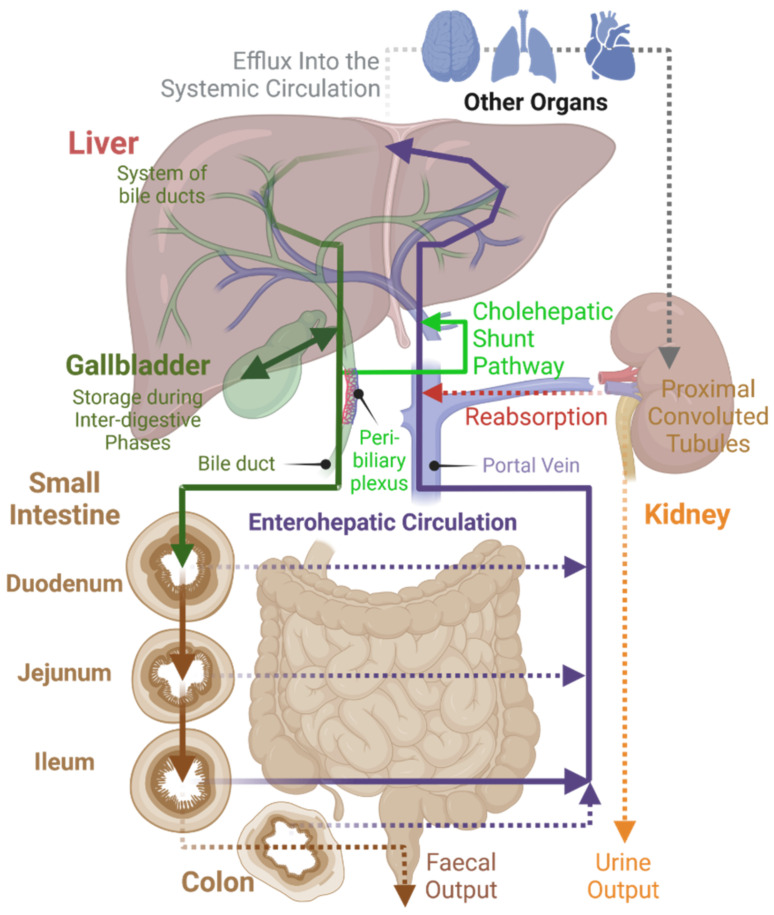
Enterohepatic circulation. Created with BioRender.com.

**Figure 4 molecules-27-02961-f004:**
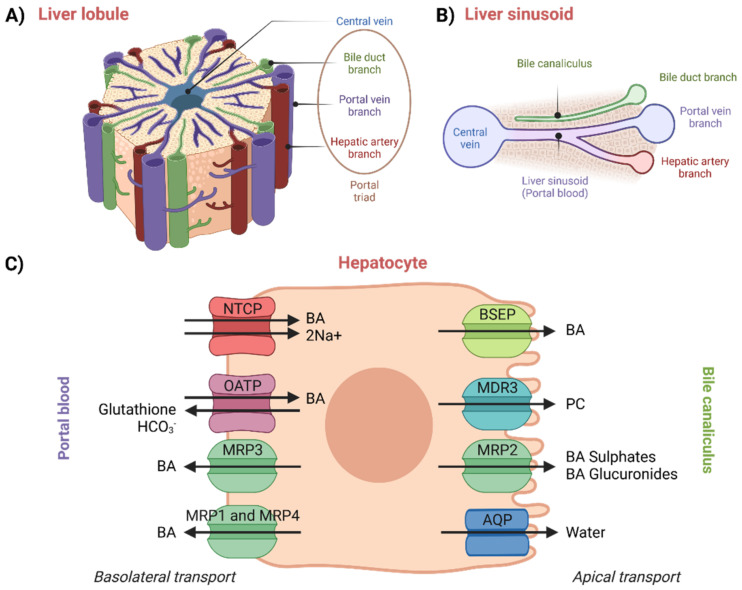
Illustration of: (**A**) liver lobule, (**B**) liver sinusoid, and (**C**) transport proteins and processes described to be taking place in hepatocytes. Created with BioRender.com.

**Figure 5 molecules-27-02961-f005:**
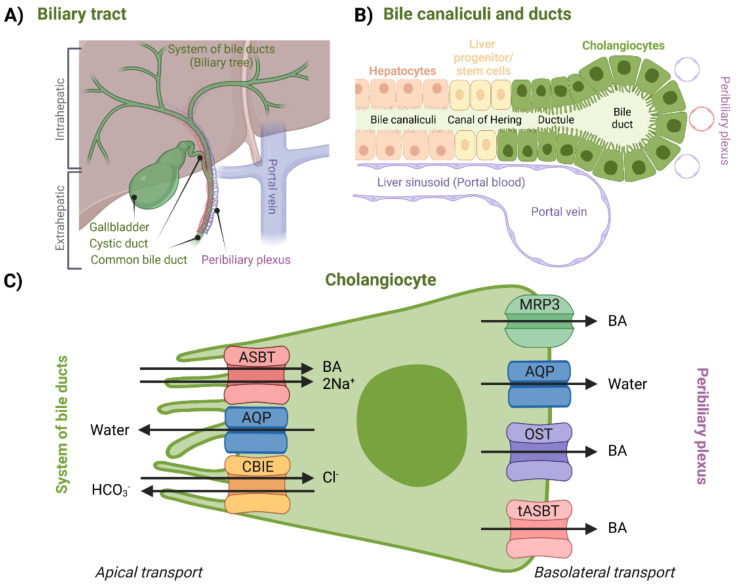
Illustration of: (**A**) biliary tract, (**B**) bile canaliculi and ducts, and (**C**) transport proteins and processes described to be taking place in cholangiocytes. Created with BioRender.com.

**Figure 6 molecules-27-02961-f006:**
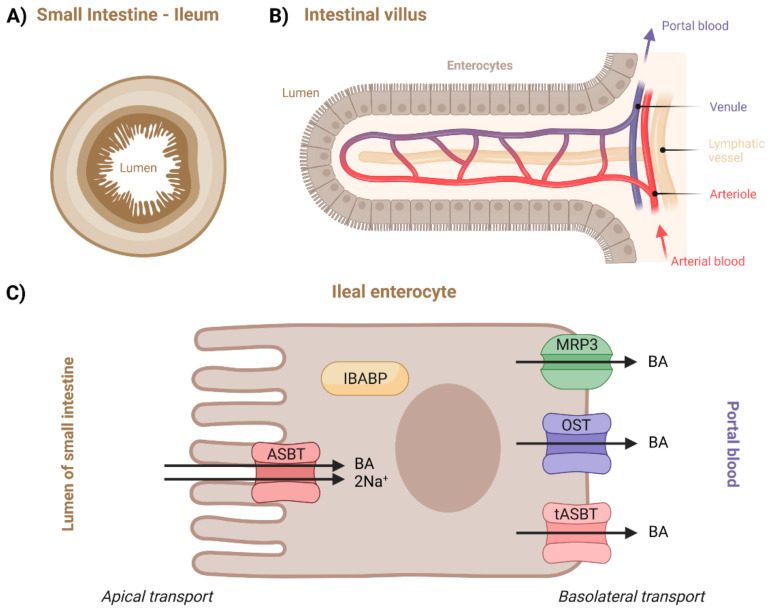
Illustration of: (**A**) ileal part of small intestine, (**B**) intestinal villus, and (**C**) transport proteins and processes described to be taking place in enterocytes. Created with BioRender.com.

**Figure 7 molecules-27-02961-f007:**
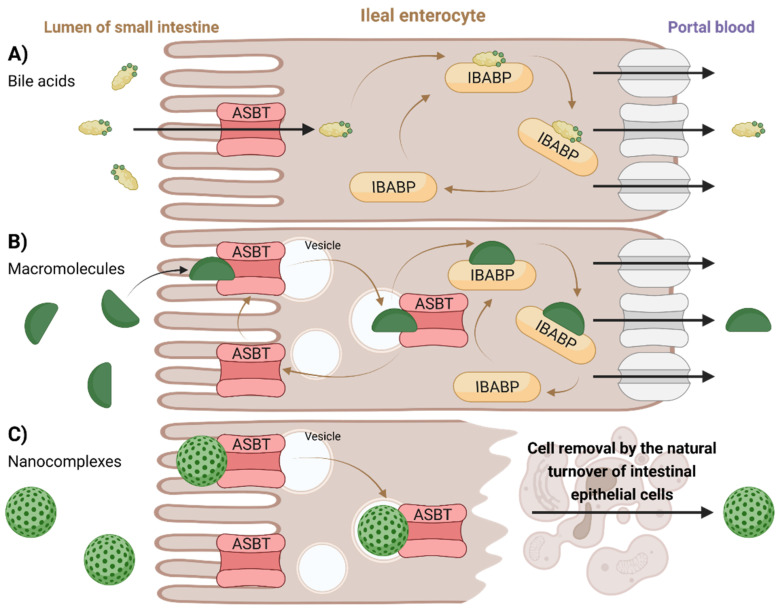
Proposed methods of intracellular transport in the ileal enterocytes. In general, BAs or BA-derived carriers are initially absorbed into the enterocytes via the ASBT protein. (**A**) Transport of BAs and small BA-conjugated drugs continues using IBABP, and later, the molecule is released into portal blood by various transporters. (**B**) BA-containing macromolecule forms a complex with ASBT protein, which is taken up in vesicles. ASBT protein is then recycled back into the membrane, while the macromolecule is released into the bloodstream by various transporters. (**C**) BA-containing nanocomplex is contained in vesicles upon ASBT-mediated uptake. The nanoparticle is then released into the bloodstream by the natural turnover process of epithelial cells. Created with BioRender.com.

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
