# Peer review of "Bile Acids Transporters of Enterohepatic Circulation for Targeted Drug Delivery"

_molecules, 2022, doi:10.3390/molecules27092961_

Round 1

Reviewer 1 Report

The review submitted by Durnik et al. is accurate and exhaustive. However, it contains too many details, sometimes redundancies, potentially overwhelming for the reader. It is important to try and shorten each section, mainly those describing experimental data in animal studies: just summarize the results leaving to the interested reader to analyze them more in depth in the original reference. Moreover, it’s necessary a thorough review of the English style for sentences often too long and for introducing the use of pronouns rather than repeated nouns (check with a mother tongue teacher).

The manuscript is then suitable for publication once these recommendations are fulfilled.

Author Response

Dear Reviewer,

Thank you for the comments. We have reduced text accordingly. We have also corrected the English style and limited the use of long sentences upon discussion with a native English speaker.

With kind regards on behalf of all the authors, Ondrej

Reviewer 2 Report

The manuscript (review) is interesting and presents an updated view on enterohepatic circulation bile acid transporters for targeted drug delivery. The text has an adequate and consistent structure with the purpose of the manuscript. The references used are sufficient and the figures contribute to a good understanding of the proposal. However, I have the following comments.

I. Major Comments:
1. After the introduction, include the "methodology" section. This section should include the criteria related to the selection of the cited papers.

2. In the introduction, I would like to include a brief paragraph on the functions of cholesterol in the body.

3. What role can the diet have in the digestive processes where the participation of bile salts is required? Briefly discuss this point. Section 1.1.

3. Figures 1, 2, 3 and 7 are very good. I suggest improving the quality of figures 4, 5 and 6. I think that the authors should include color and maintain a format similar to figure 3.

II. Minor comments:
1. Improve the wording of the objective of the review study.
2. In figures 4, 5 and 6, delete "Overview". Improve the titles of those figures.

Author Response

Dear Reviewer,

Thank you for the comments. We have improved the manuscript based on your recommendations. Moreover, we have also prepared the Figure 7 to match the style and to further improve the quality of our manuscript. We have also added detailed schematic description of the transport pathways in figures 4-6.   

With kind regards on behalf of all the authors, Ondrej

Reviewer 3 Report

Very well written summary of BA synthesis, secretion, and transport.

Line 185 the authors state that “A small proportion of the BAs is deconjugated 185 as they pass down toward the ileum.” This is not correct – the % unconjugated BA increases as we move down the small intestine.

Author Response

Dear Reviewer,

Thank you for the comment. We certainly agree that the percentage of unconjugated BA increases as they move down the small intestine and this is, in fact, affirmed in our sentence. Yet, we have changed the wording to highlight the importance of this mechanism (removed word “small”): “a proportion of the BAs is deconjugated as they pass down the small intestine toward the ileum”.

With kind regards on behalf of all the authors, Ondrej